# Investigation on Intestinal Proteins and Drug Metabolizing Enzymes in Simulated Microgravity Rats by a Proteomics Method

**DOI:** 10.3390/molecules25194391

**Published:** 2020-09-24

**Authors:** Huayan Liu, Jingjing Guo, Yujuan Li, Yushi Zhang, Jiaping Wang, Jianyi Gao, Yulin Deng, Yongzhi Li

**Affiliations:** 1School of Life Science, Beijing Institute of Technology, No.5 Zhongguancun South Street, Haidian District, Beijing 100081, China; lhy7881230@163.com (H.L.); jingjingguo926@163.com (J.G.); deng@bit.edu.cn (Y.D.); 2Institute of Chinese Materia Medica, No.16 Dongzhimen Neinan Street, Dongcheng District, Beijing 100081, China; yszhang@icmm.ac.cn; 3Astronaut Research and Training Center of China, No.109 Youyi Road, Haidian District, Beijing 100094, China; wangjiaping_1113@163.com (J.W.); gaojianyi507@sina.com (J.G.)

**Keywords:** simulated microgravity, intestinal mucosa, proteomics, metabolic pathways, intestinal drug metabolic enzymes

## Abstract

The present study aimed to investigate the change of intestinal mucosa proteins, especially the alteration of intestinal drug metabolizing enzymes (IDMEs) following 14-day simulated microgravity. Morey–Holton tail-suspension analog was used to simulate microgravity. Intestinal mucosa proteins of rats were determined by label-free quantitative proteomic strategy. A total of 335 differentially expressed proteins (DEPs) were identified, 190 DEPs were upregulated, and 145 DEPs were downregulated. According to bioinformatic analysis, most of DEPs exhibited hydrolase, oxidoreductase, transferase, ligase, or lyase catalytic activity. DEPs were mainly enriched in metabolic pathways, including metabolism of amino acid, glucose, and carbon. Moreover, 11 of DEPs were involved in exogenous drug and xenobiotics metabolism. Owing to the importance of IDMEs for the efficacy and safety of oral drugs, the expression of cytochrome P450 1A2 (CYP1A2), CYP2D1, CYP3A2, CYP2E1, alcohol dehydrogenase 1 (ADH1), and glutathione S-transferase mu 5 (GSTM5) in rat intestine mucosa was determined by Western-blot. The activity of ADH, aldehyde dehydrogenase (ALDH) and GST was evaluated. Compared with control rats, the expression of CYP1A2, CYP2D1, CYP3A2, and ADH1 in the simulated microgravity (SMG) group of rats were dramatically decreased by 33.16%, 21.93%, 48.49%, and 22.83%, respectively. GSTM5 was significantly upregulated by 53.14% and CYP2E1 expression did not show a dramatical change in SMG group rats. Moreover, 14-day SMG reduced ADH activity, while ALDH and GST activities was not altered remarkably. It could be concluded that SMG dramatically affected the expression and activity of some IDMEs, which might alter the efficacy or safety of their substrate drugs under microgravity. The present study provided some preliminary information on IDMEs under microgravity. It revealed the potential effect of SMG on intestinal metabolism, which may be helpful to understand the intestinal health of astronauts and medication use.

## 1. Introduction

Complex space environment, including microgravity (MG), strong radiation, and high noise, could lead to multi-system damage of organisms. It was reported that microgravity or simulated microgravity (SMG) might cause muscle atrophy [1,2], bone loss [3,4], cardiovascular dysfunction [5,6], nervous system damage [7,8], immune function attenuation [9,10], intestinal barrier destruction [11,12,13], and so on. Therefore, looking for counter measures against the physio-pathological changes induced by microgravity has always been an important research topic [14]. So far, besides physical training, lower body negative pressure and saline supplementation, drugs were often used to prevent or treat the body injury induced by microgravity. For example, promethazine has the effect of alleviating symptoms of space motion sickness and was used aboard the U.S. Space Shuttle [15]. It has been reported that more than 70% of crew members used zolpidem or temazepam as a sleep aids during both short/long duration spaceflight missions and International Space Station missions [16,17]. It is better to assess the effectiveness and safety of drugs during space flight to maintain astronaut health and performance [18].

The small intestine is one of the important organs of the body. It exhibits the functions of digestion, absorption, and metabolism of various substances. Proteins, glucose, and lipids in food can be digested and utilized to varying degrees in the small intestine, and provide nutrition and energy for the body [19,20]. It is known that the intestine is also the main place for absorption of oral drugs [21]. Some intestinal drug metabolizing enzymes (IDMEs) are present in the small intestine [22], which play important roles in the metabolism and/or detoxification of drugs and xenobiotics [23]. IDMEs could be classified into two main types. One type is oxidative enzymes that mainly mediates phase I reactions, whereas the other type is conjugative binding enzymes [24]. Phase I IDMEs include cytochrome P450 (CYP450), flavin monooxygenases (FMO), aldehyde oxidases (AOX), alcohol dehydrogenases (ADH), aldehyde dehydrogenases (ALDH), hydrolases, and so on. Phase II IDMEs comprise UDP-glucuronosyltransferases (UGTs), glutathione S-transferases (GST), N-acetyltransferase, and others [25]. In summary, when drugs are bio-transformed by IDMEs, their efficacy and safety may be changed.

Currently, few studies have been focused on the effects of MG on enzymes in the intestine. Reports have only shown that contents of some intestinal metabolic enzymes (such as leucine aminopeptidase, acid phosphatase, adenosine triphosphatase, and glucose-6-phosphatase) were significantly increased in the rat digestive tract during 7 d, 13 d, or 18 d space flights (in the Soviet biosatellite Cosmos 1667, 1887, and 1129) [26,27,28]. The effects of MG or SMG on IDMEs remain unexplored. Reports of hepatic metabolic enzymes under MG or SMG are available. It has been found that the CYP450 content and the activity of CYP450 dependent enzymes decreased in the liver microsomes of 14 d space flying Wistar rats (in Cosmos 1887) [29]. CYP2C29, CYP2E1, and CYP1A2 contents were significantly increased in the liver of mice exposed to 30 d space flight [30]. Chen, et al. found that 3 d and 14 d SMG had a significant effect on the expressions of CYP1A2, CYP2C11, CYP2D1, and CYP3A2 in rat liver [31]. It is known that some subtypes of drug metabolizing enzymes in the intestine are like those in the liver. So, it could be speculated that IDMEs might also be affected by MG as well. The alterations of IDMEs may change the pharmacokinetics and/or pharmacodynamics of drugs used by astronauts, and then may affect the drug efficacy and safety [32]. For example, the hypnotic drug zolpidem and analgesics alprazolam used by astronauts could be metabolized by CYP450 in the small intestine; thus, their efficacy were weakened [33,34,35]. At present, astronauts used medications according to the terrestrial medical practices, but it is not known whether the drugs will act on the body in spaceflight as the same way on Earth or not. Many physiological changes caused by the space environment have been not fully considered, such as the changed motility of gastrointestinal tract on drug absorption and the effect of changed drug metabolizing enzymes on drug metabolism [32,36]. Obviously, further research on IDMEs under MG or SMG condition is essentially needed. Due to the limitation of low frequency of space flights, MG analog on ground is widely used for scientific research, such as rat tail-suspension [37], rabbit head-down rest [38], and bed-rest of human [39]. Among them, Morey–Holton tail-suspension analog is recognized as a well-accepted ground-based spaceflight analog by National Aeronautics and Space Administration (NASA) [40].

The present study aimed at investigating the whole change of jejunum mucosa proteins induced by 14 d SMG based on a label-free quantitative proteomics method, and especially focused on change of IDMEs. The expression of some IDEMs (CYP1A2, CYP2D1, CYP2E1, CYP3A2, ADH1, GSTM5), and the activities of ADH, ALDH, and GST in rats were determined. The results might be helpful to understand the response of intestinal mucosa protein to 14 d SMG, and disclose the change of IDMEs under SMG condition.

## 2. Results and Discussion

Before proteomics study, the histomorphology of rat intestinal mucosa under 14 d SMG was observed by hematoxylin–eosin (HE) staining, shown in Appendix A. It was found that the intestinal villi of 14 d SMG rats was abnormal in morphology with signs of necrosis and exfoliation, suggesting that 14 d SMG induced the intestinal mucosal barrier damage. The crypts were swollen and the number of goblet cells in the crypt was reduced. Crypts mainly comprises columnar cells, goblet cells, and Paneth cells, which can synthesize and secrete a variety of enzymes involved in the digestion and metabolism of glucose, proteins, amino acids, lipids, and other substances. The decreased number of goblet cells and damaged crypt structure under SMG might impair the digestive and metabolic functions of the small intestine. In order to screen the overall proteins associated with metabolism, the proteomic method was employed to further investigate the change of rat jejunum mucosa proteins induced by 14 d SMG.

### 2.1. Proteomic Analysis

In total, 1826 proteins were identified in rat jejunum mucosa, which was shown in the volcano plot in Figure 1a. Since the cutoff values of fold change for upregulation and downregulation were set at >2 and <0.5, respectively, and the *p*-value of *t*-test was lower than 0.05, 335 proteins were identified as differentially expressed proteins (DEPs), shown in Figure 1b. Information of all DEPs is shown in Appendix A. Among them, the expression of 190 proteins increased and the expression of 145 proteins decreased. The mass spectrometry proteomics data have been deposited to the ProteomeXchange Consortium (http://proteomecentral.proteomexchange.org) via the iProX partner repository [41] with the dataset identifier PXD021522.

Mass spectrometry results indicated that ADH1 was downregulated and GSTM5 was upregulated under 14 d SMG. The Western-blot analysis has successfully validated these results. The result of Western-blot was shown in Figure 1c. Compared with the control group (CON, normal gravity), the expression of ADH1 decreased and GSTM5 increased under 14 d SMG. It was consistent with the results of mass spectrometry (MS).

Compared with 7 d SMG experiments [11], 159 DEPs in 14 d SMG were the same as those in 7 d SMG. The number of DEPs in intestinal mucosa of 14 d SMG treated rats was less than that of in 7 d SMG rats. The proportion of upregulated proteins to all DEPs under 14 d SMG condition was decreased compared with 7 d SMG treatment. Significantly changed numbers and expression tendency of DEPs may be associated with the different SMG durations. Comparison results between DEPs of 7 d and 14 d SMG were shown in Figure 2. From 7 d to 14 d SMG treatment, 558 DEPs in rat intestinal mucosa showed the tendency back to normal levels. Moreover, 111 DEPs were continuously upregulated, and 7 DEPs were continuously downregulated. For example, the expression of metabolic-related proteins, such as hexokinase (HK) and GSTs, continued to increase, which suggested that metabolism mediated by HK and GSTs might be sensitive to simulated microgravity. Different SMG duration led to distinct effects on expression of proteins in rat intestinal mucosa.

#### 2.1.1. Gene-Ontology Classification

A total of 335 DEPs were analyzed using protein annotation through evolutionary relationship (PANTHER) classification system (version 15.0, http://www.pantherdb.org/, Thomas lab at the University of Southern California, Los Angeles, CA, USA) for Gene-Ontology (GO) classification, including biological process (BP) and molecular function (MF) annotation. GO classification results were shown in Figure 3a,b. In order to show more informative results, the top two classifications of BP and MF with more details were further expanded in pie figures, according to gene counts (Figure 3). More information on the most important biological processes (metabolic process and biological regulation), and molecular functions (catalytic activity and binding) were shown in these figures.

In BP annotation, DEPs were remarkably enriched in metabolic process, biological regulation, cellular component organization, or biogenesis, and others. Moreover, 122 DEPs were included in the metabolic processes, accounting for 36.4% of the total DEPs. After the metabolic processes were further analyzed, it has been found that the DEPs were involved in the metabolism of amino acids, glucose, lipids, and proteins. Meanwhile, the metabolism of drugs and other xenobiotics was also found in the metabolic processes. Secondly, 71 DEPs were involved in biological regulation, including regulation of metabolic processes, homeostatic processes, signaling, and other processes. The regulation of primary metabolic process of macromolecule and nitrogen compound, the anatomical structure and chemical homeostasis of the small intestine were mainly affected by 14 d SMG. In addition, cellular component organization or biogenesis also could be affected by SMG condition, such as ribonucleoprotein complex biogenesis and organelle, protein-containing complex subunit, membrane organization.

In terms of MF annotation, the top two affected by 14 d SMG were catalytic function and binding function, following molecular function regulation, transporter, and others. Moreover, 127 DEPs had catalytic activity, including hydrolase, transferase, oxidoreductase, isomerase, ligase, and lyase activity. Among them, 85 DEPs, accounting for 66.9% of 127 DEPs, participated in metabolic processes of amino acids, glucose, drugs, and other xenobiotics as catalytic enzymes. The rest of the DEPs were mainly involved in hydrolysis of acid anhydrides and ester bonds, transfer of phosphorus-containing groups, monooxygenation, etc. In addition, 113 DEPs had binding function and could bind to protein, protein-containing complex, carbohydrates, and their derivatives, lipids, drugs, or small molecules. Nearly half of the DEPs with binding function also had catalytic function, and participated in the metabolism of amino acids, glucose, drugs, and other xenobiotics. 

According to the results of BP and MF, it could be concluded that SMG significantly affected the metabolic process of amino acids, glucose, drugs, and other xenobiotics in the small intestine. It may further have an influence on the energy supply to the body, drug efficacy, and intestinal homeostasis.

#### 2.1.2. Kyoto Encyclopedia of Genes and Genomes (KEGG) Pathway Analysis

Protein-protein interaction network of all DEPs were analyzed by the Search Tool for the Retrieval of Interacting Genes/Proteins (STRING, version 11.0, Hinxton, Cambridgeshire, UK), shown in Appendix A. Network nodes represent proteins, and edges represent protein-protein associations. The DEPs formed a complex protein-protein interaction network, and some of them could directly interact with several other proteins. Direct or indirect interaction of DEPs may affect potentially some pathways. In order to further disclose the pathways affected by 14 d SMG, KEGG analysis was performed by Database for Annotation, Visualization and Integrated Discovery(DAVID, version 6.8, https://david.ncifcrf.gov/, Laboratory of Human Retrovirology and Immunoinformatics, Frederick, MD, USA).

KEGG pathway analysis of DEPs could understand how many pathways could be affected by SMG in intestinal mucosa. The clustering results were shown in the Figure 3c. The results showed that most DEPs were enriched in metabolism pathway, ribosome, biosynthesis of antibiotics, and other pathways. Further analysis of metabolic pathways revealed that it mainly included the metabolism of various types of glycans, monosaccharides, amino acids, fatty acids, vitamins, sulfur, pyrimidine, purine, etc. In addition, the metabolism of carbon, amino acids (beta-alanine, valine, leucine, and isoleucine), drugs and xenobiotics, glycolysis, and glycogenesis were also affected by SMG. The KEGG results were like those of BP and MF, which confirmed that DEPs were mainly enriched in metabolism pathways. SMG would significantly affect the metabolism of amino acids, glucose, drugs, and xenobiotics in the small intestine.

Our previous results indicated that 7 d SMG mainly affected cell-cell adhesion, fatty acids metabolism, oxidative stress, and others [11]. While 14 d SMG mainly influenced glucose metabolism, drugs and xenobiotics metabolism, amino acids metabolism and defense mechanisms against bacteria. Comparing the results of 7 d and 14 d SMG, it showed that different SMG durations had different influence on the small intestinal mucosa of rats. The study has shown that lipid metabolism in the liver of mice flown aboard (for 13.5 d) the Space Transportation System-135 was also affected by microgravity, resulting in increased triglyceride storage and loss of retinols [42]. However, no significant effect on lipid metabolism was observed in the intestine of 7 d and 14 d SMG rats. Combining GO and KEGG results, the present study focused on glucose metabolism, amino acids metabolism, drugs, and xenobiotics metabolism for further analysis.

### 2.2. Metabolism-Related DEPs

#### 2.2.1. Glucose Metabolism

The present results of proteomics indicated that 9 DEPs were involved in glucose metabolism (shown in Table 1). The expressions of the aldehyde dehydrogenase family 1A3 (ALDH1A3), cytosolic phosphoenolpyruvate carboxy kinase (PCK1), hexokinase-2 (HK2), HK1, mitochondrial aldehyde dehydrogenase (ALDH2), and aldehyde dehydrogenase family 3-member A2 (ALDH3A2) increased 2.17 to 3.37 times, respectively. ATP-dependent 6-phosphofructokinase (PFK), fructose-bisphosphate aldolase B (ALDOB) and ADH1 were downregulated 0.12 to 0.49 times of the CON group. HK, PFK, and PCK were involved in the key steps of glycolysis and gluconeogenesis processes, as shown in Figure 4.

HK1 and HK2 participate in phosphorylation of α- and β-d-glucose to α- and β-d-glucose-6-phosphate [43], promoting the glycolysis process. Furthermore, PFK can phosphorylate β-d-fructose-6-phosphate to fructose-1,6-bisphosphate [44]. Moreover, α-d-glucose-6-phosphate, β-d-glucose-6-phosphate, and β-d-fructose-6-phosphate are interconverted, further enter the pentose phosphate pathway. Pentose phosphate pathway is a major pathway of glucose metabolism. In addition to providing energy, it also provides a variety of raw materials for metabolism. Energy supply affects cellular proliferation, development, differentiation, and a variety of other basic life activities. It has been reported that the expression of HK was decreased in the liver of 11 d hindlimb-unloading mouse [45], which meant SMG led to a decrease in glucose metabolism in the liver. The expression of PFK was significantly increased in 3 d SMG rat livers [46]. The study also showed that the expression of HK and glycolytic activity in the muscle of 21 d tail-suspension rats were significantly increased, and the increase of glycolysis was used to supplement the energy deficiency caused by the decrease of lipid metabolism [47]. In the present study, the expression of HK1 and HK2 was upregulated, and PFK was downregulated in the small intestine, which was opposite to the liver. Combining the literature and ours results, it can be indicated that SMG may change the glucose metabolism in the small intestine; thus, changing the energy supply for the basic life activities.

PCK1 is a key enzyme in the metabolic network of gluconeogenesis, energy metabolism, and tricarboxylic acid cycle [48]. When the body is at low glucose levels, it can catalyze pyruvate to phosphoenolpyruvate, the rate-limiting step from lactate and other precursors derived from the citric acid cycle to glucose [49]. Rodent models also demonstrated that over-expression of PCK1 could result in type 2 diabetes development [50]. The expression of PCK1 in the small intestine under MG and SMG has not been reported, but it increased significantly in the present study. Under normal circumstances, glycolysis and gluconeogenesis are the main processes of the decomposition and synthesis of glucose in vivo, which regulate each other to maintain the balance of glucose metabolism in the body. Under SMG conditions, the expression of HK1, HK2, and PCK all increased significantly, indicating that both glycolysis and gluconeogenesis in the small intestine were affected, which may lead to disruption of the balance of glucose metabolism in the small intestine.

The study has pointed out that ADH1 plays a new role in human glucose metabolism, which can catalyze the conversion of erythrose to erythritol [51]. The average glycemic index and insulin index of erythritol are lower than those of general glycols. Therefore, the effect of erythritol on blood glucose is smaller. Under 14 d SMG, the decreased expression of ADH1 will reduce the synthesis of erythritol, which may lead to elevated blood glucose. At the same time, as a marker of β-cell dedifferentiation, the increase of ALDH1A3 expression will lead to the decrease of insulin secretion [52]. It can be speculated that SMG may induce polydipsia, polyphagia, polyuria, weight loss, and other symptoms if the intestinal glucose metabolism was seriously disturbed [53].

In conclusion, the SMG will significantly affect the glucose metabolism in the small intestine of rats, and may lead to disorder of glucose metabolism.

#### 2.2.2. Amino Acids Metabolism

Present results of proteomics showed that 11 DEPs were related to metabolism of amino acids (shown in Table 2). The expressions of 4 ALDH family proteins, including ALDH6A1, ALDH1A3, ALDH2, and ALDH3A2, were upregulated. Meanwhile, compared with CON group, mitochondrial 4-aminobutyrate aminotransferase (ABAT), mitochondrial propionyl-CoA carboxylase beta chain (PCCB), peroxisomal bifunctional enzyme (EHHADH), membrane primary amine oxidase (AOC3), Beta-Ala-His dipeptidase (CNDP1), and mitochondrial branched-chain-amino-acid aminotransferase (BCAT2) were overexpressed in the small intestine of SMG group rats. In addition, only the expression of Aldehyde oxidase 2 (AOX2) decreased significantly. The above 11 DEPs were involved in the metabolism of essential amino acids beta-alanine, l-leucine, l-isoleucine, and l-valine, as shown in Figure 5.

ALDHs were important in detoxification of aldehydes and metabolism of amino acids. ALDH1A3, ALDH2, and ALDH3A2 can catalyze the conversion of β-aminopropion-aldehyde to β-alanine, ALDH6A1 can transform malonate-semialdehyde into Acetyl-CoA, which were related to the metabolism of β-alanine. Moreover, they also participate in the metabolism of l-valine. Mutations in genes encoding ALDHs could cause metabolic disorders, including alcohol flush reaction (ALDH2), Sjögren–Larsson syndrome (ALDH3A2), and methylmalonic aciduria (ALDH6A1) [54]. The activity of ALDH was also determined in this study, and the result was shown in Table 3. ALDH showed a slight increase in the small intestine of SMG group rats without significant difference. Few studies have been found on the changes of amino acid metabolism in the small intestine under MG or SMG. Research has showed that the content of valine decreased significantly in peripheral blood mononuclear cells under SMG [55]. The high expression and activity of ALDH in the small intestine suggests that SMG may promote the metabolism of β-alanine and l-valine in the small intestine; thus, reducing the content of amino acids entering other tissues. 

ABAT is a member of class-III pyridoxal-phosphate-dependent aminotransferase family, and it is one of the key enzymes in the decomposition of γ-amino butyric acid [56], which is a signaling molecule and can modulate the cell cycle and apoptosis [57]. PCCB is a subunit of the biotin-dependent propionyl-CoA carboxylase (PCC), which can catalyze the decomposition of branched-chain amino acids, such as isoleucine, threonine, methionine, and valine [58]. CNDP1 can hydrolyze not only the beta-Ala-His dipeptide preferentially, but also other dipeptides, such as homocarnosine [59]. BCAT2 is the key enzyme that catalyzes the first reaction in the catabolism of l-leucine, l-isoleucine and l-valine [60], converting them into 4-Methyl-2-oxopentanoate, (S)-3-Methyl-2-oxopentanoate, and 3-Methyl-2-oxobutanoate, respectively. The expression of these enzymes was significantly increased in the small intestine of rats in the SMG group, which may promote the catabolism of a variety of amino acids in the small intestine.

Intestinal amino acid catabolism is necessary for maintaining intestinal mucosal functions and integrity. It not only provides the required energy for ATP-dependent physiological processes in the small intestine [61], but also plays an important role in intestinal epithelial cells proliferation, differentiation and repair [62], regulation of local and systemic blood flow [63], and antioxidant damage [64]. Therefore, increased amino acid metabolism in the small intestine of rats under SMG conditions is beneficial to maintain the small intestinal mucosal homeostasis. However, if the essential amino acids in food were over metabolized when they pass through the intestinal mucosa, it may alter the amount of amino acids entering the portal circulation and supplying to tissues other than the intestine, resulting in the reduction of the utilization of amino acids under SMG [65]. 

#### 2.2.3. Drugs and Xenobiotics Metabolism

Present results of proteomics indicated that 11 DEPs were involved in drugs and xenobiotics metabolism, listed in Table 4. The 14 d SMG mainly altered the expression of GST enzyme system (GSTM5, MGST1, GSTA2, GSTA4, and GSTA5) and ADH enzyme system (ADH1 and ALDH1A3) in the small intestine of rats. The expression of FMO5, glutathione peroxidase 1 (GPX1), aminopeptidase N (APN), aldehyde oxidase 2 (AOX2) were also affected.

##### SMG Effect on ADH Enzyme System (ADHs) 

The most common substrate of ADHs is ethanol. It also participates in the metabolism of xenobiotic alcohols and aldehydes, steroids, biogenic amines, lipid peroxidation products, and ω-hydroxy fatty acids. When ADHs metabolize alcohol and aldehydes, it doesn’t produce toxic radicals like the cytochrome P450 system, so ADHs was considered to involve in the general defense against alcohols and aldehydes [66]. Under normal conditions, ethanol in the body is mainly metabolized by the ADHs. However, the non-ethanol dehydrogenase system, whose key enzyme is CYP2E1, is also hyperactive when the ADHs is insufficient or the ethanol content is too high. In addition to reactive oxygen species, the non-ethanol dehydrogenase system will produce the toxin acetaldehyde, which causes oxidative damage to the body [67]. Activity assay showed that compared with CON group, the activity of ADH in the small intestine of SMG group rats was significantly reduced by 38.32% (Table 3). Therefore, the significantly decreased expression of ADH1 and activity of ADH may promote the alcohol metabolize in non-ADH system in intestine, leading to intestinal injury.

Because ADHs play an important role in the metabolism of retinoid, dopamine, and detoxification of aldehyde, it may be associated with the pathogenesis of neurodegenerative diseases [68,69]. There was a lack of mRNA encoding the corresponding ADHs in the brain, ADHs and ALDH1 were mainly expressed in the digestive tract. Low expression and activity of ADHs in the gastrointestinal tract may led to toxic substances, such as aldehydes, reaching central system neurons through blood circulation [70]. 

##### SMG Effect on GST Enzyme System (GSTs)

GSTs are the key enzymes that catalyze the initial step in glutathione binding reactions. At present, there are 5 types of cytoplasmic isozymes and microsomal GST found in human GSTs, among which cytoplasmic isozyme subtypes α, π, μ, and θ are abundant in the human body [71]. GSTs can catalyze the binding reaction of nucleophilic glutathione with various kinds of electrophilic exogenous chemicals, forming more soluble and nontoxic derivatives, and make it easy to excrete or to further metabolize [72]. After being metabolized by CYP450, many drugs will form some bioactive intermediates, which can covalently combine with important components of cell biomacromolecules, causing damage to the body. If glutathione was combined with these intermediates, this covalent binding can be prevented, achieving the effect of detoxification [73]. GSTs were also involved in the metabolism of most xenobiotics, such as benzopyrene, naphthalene, trichloroethylene [74,75,76,77,78,79], by catalyzing the combination of glucuronic acid and lipophilic aglycon in xenobiotics. Because the small intestine has a high metabolic rate and self-renewal rate, many reactive oxygen species (ROS) are produced in the small intestine. ROS accumulated in the absence of timely elimination can attack the cell membrane system and mitochondrial DNA [80]. Glutathione, involved in the elimination of ROS, is essential to prevent oxidative stress and protect cell membranes and DNA from ROS damage [81]. Studies have also shown that the level of GST expression in vivo determines the sensitivity of cells to some toxic chemicals [82]. GSTs play a role in detoxification and anti-oxidation functions when the organism is exposed to radiation, microgravity, and other conditions, to protect the organism. 

After 14 d SMG, the expression of GSTM5 and MGST1 was upregulated, while the expression of GSTA4, GSTA2, and GSTA5 was downregulated. The total enzyme activity of GST showed a downward trend without significant difference, shown in Table 3. The study has also shown that a 9 d space flight in SLS-1 induced intestinal GST activity, but there were no significant changes of intestinal GST activity in SLS-2 (a 14-day space flight) [83], suggesting that the effect of microgravity on GST activity was related to the flight duration. 

The changed activity and expression of GST indicated that SMG conditions may affect the function of GST in the intestine, including detoxification, xenobiotics metabolism, and ROS elimination.

##### SMG Effect on FMO and AOX

FMO is an important microsomal metabolic enzyme of drugs and chemical xenobiotics, which can catalyze the oxidation of compounds and drugs containing nitrogen, sulfur, phosphorus, selenium, and other affinity heteroatoms [84]. AOX is a broad substrate-specific oxidase that oxidizes aldehydes, nitrogen-containing, and oxygen-containing heterocyclic compounds. It plays an important role in phase I metabolism of drugs and exogenous substances [85]. The upregulation of FMO5 expression and downregulation of AOX2 expression induced by SMG conditions may affect the metabolism of its substrate drugs in the small intestine, thus, affecting the efficacy and safety.

In a previous study, GSTM5 and MGST1 were found to be significantly increased in 7 d SMG rats [11], while they showed much higher increase in 14 d SMG rats. Further, GSTA2, GSTA4, GSTA5, FMO, and AOX were only found in 14 d SMG rats. Different SMG durations showed a different effect on IDMEs, which may lead to different drug metabolism outcomes. 

### 2.3. Determination of Drug Metabolism CYP450

CYP450 is a family of enzymes consisting of a series of oxidases containing heme coenzymes, which is the main metabolic enzyme of exogenous substances, such as drugs, poisons, and environmental carcinogens [86]. Among all enzymes in CYP450 family, CYP1, CYP2, and CYP3 are main subtypes involved in drug metabolism [87]. More than 90% of drugs are metabolized by CYP450, which mainly exist in the liver and small intestine. MS did not detect the differential expression of CYP450 in present study. Because of their important roles in drug metabolism, we focused on the changes of them under SMG, and defeminated the expression of CYP1A2, 2D1, 2E1, and 3A2 in the rat small intestine. 

The CYP3A is the most important member of the CYP450 family, because it is abundant in the liver and intestine, and metabolizes a wide range of clinical drugs [88]. There are four subtypes of CYP3A in the human body, including 3A3, 3A4, 3A5, and 3A7 [89]. Among them, CYP3A4 is the main isoenzyme, which corresponds to rat CYP3A2 [90], and mainly distributes in the liver and small intestine. CYP3A4 can affect the metabolism of drugs, and cause many drug interactions [91]. CYP2D consists of CYP2D6, CYP2D7, and CYP2D8. Only CYP2D6 can be expressed in human tissues, such as liver and intestine [92]. It corresponds to rat CYP2D1 [89,93]. CYP2D6 can catalyze the metabolism of more than 30 drugs, such as amitriptyline, fluoxetine, propafenone, metoprolol, and propranolol [94]. CYP1A2 is mainly involved in the metabolism of aromatic amines, heterocyclic amines, and some halogenated hydrocarbons, and it is also an important isozyme for the metabolism of certain xanthine drugs [95]. Human CYP2E1 is mainly distributed in the adult liver, and it is also found in some extrahepatic tissues, such as intestines and lungs. The substrates of CYP2E1 are mostly pre carcinogens, toxins, and a few drugs [96]. The expression of CYP1A2, CYP2D1, CYP2E1, and CYP3A2 were determined by Western-blot, the results were shown in Figure 6. Compared with the CON group, the expression of CYP1A2, CYP2D1, and CYP3A2 significantly decreased by 33.16%, 21.93%, and 48.49% under 14 d SMG, respectively. While, there was no significant change in CYP2E1. 

Astronauts in the spaceflight will take some drugs to protect the physiological and psychological damage caused by complex space environment, to keep health and improve performance efficiency. For example, antibiotics (azithromycin, cefalexin, ciprofloxacin), analgesics (acetaminophen, aspirin, ibuprofen), cardiovascular drugs (propranolol, verapamil), antihistamines (promethazine, diphenhydramine), and gastrointestinal drugs (omeprazole). They are all standby drugs on the U.S. space station. Some of these oral drugs can be metabolized by CYP450 in the human small intestine, such as ciprofloxacin, promethazine and omeprazole, whose efficacy and safety probably be affected.

Ciprofloxacin is one of a new generation of fluorinated quinolones, and it was used in the U.S. space station as antibiotic against respiratory infections [97]. Ciprofloxacin is a substrate and inhibitor of CYP1A2. The expression of CYP1A2 in rat small intestinal mucosa was significantly downregulated under 14 d SMG, and this may decrease ciprofloxacin metabolism. It may result in ciprofloxacin accumulation in vivo, inducing side effects, including mild gastrointestinal irritation or discomfort, nausea, heartburn, loss of appetite, mild nervous system reactions, etc. [98]. Promethazine is an antihistamine, a drug used to treat space motion sickness [99]. When it is absorbed in the gastrointestinal tract, it can be metabolized by CYP2D6 (corresponding to rat CYP2D1), into ring-hydroxylation, S-oxidation and N-demethylation, and then becomes ineffective [100,101]. It has been reported that the low bioavailability of oral promethazine may be related to its high metabolism in the small intestine [102]. According to the results of the present study, the decreased expression of CYP2D1 in the small intestine may help improve the bioavailability of oral promethazine under SMG. Omeprazole is an acid suppressant used to treat severe stomachache or gastric ulcers [103]. Intestinal CYP3A2 mediates the first-pass metabolism of oral omeprazole [104]. Decreased CYP3A2 under SMG may reduce the metabolism of omeprazole in the small intestine, and then may change its plasma concentration level. 

It could be concluded that SMG can change the expression of CYP1A2, CYP2D1, and CYP3A2 in the intestine, thus, affecting the metabolism of some aerospace drugs (the substrates of these intestinal metabolic enzymes) in the small intestine. As far as we know, there is no report about the effect of MG or SMG on the expression of CYP450 in small intestine. The present research may be helpful to further study drug intestine metabolism during spaceflight. It should be noted that the Morey–Holton model is a ground analog to simulate gravity, it needs further research to verify the real microgravity effect on IDMEs. We tried to determine activity of CYP450 enzymes in the small intestine; however, it could not be carried out successfully. It may result from the relatively weak activities of CYP450 enzymes in the small intestine under SMG. More effort should be made in further research.

## 3. Materials and Methods 

### 3.1. Animals Treatment and Samples Collection

The present study complied with the Guide for the Care and Use of Laboratory Animals published by the National Institutes of Health (NIH publication No.85-23, revised in 1996), and all animal experiments were approved by the Beijing Institute of Technology Animal Care and Use Committee (Beijing, China). The approval number was 2018-0003-M-2020009 (approved in April 2020). A total of 12 male Sprague–Dawley (SD) rats (220 ± 20 g, 10-week old, Specific pathogen Free (SPF) degree) were randomly divided into two groups: CON and SMG groups. All rats were kept in the same environment with the temperature of 24 ± 1 °C and the humidity of 55 ± 5%. The rats in the SMG group were tail-suspended to make their hind limbs off the ground and produce a 30° head-down tilt based on the Morey–Holton model for 14 days (14 d) [37]. Rats in both groups had freedom of movement in the cage, and had free access to food and water. After 14 d tail-suspension, all rats were feed-deprived overnight, and anesthetized with chloral hydrate at a dose of 350 mg/kg. Rat jejunum was cut out, about 1 cm jejunum was fixed in 4% paraformaldehyde over 24 h (Solarbio, Beijing, China) for histomorphology. Jejunum mucosa was collected and stored at −80 °C for further study.

### 3.2. Histomorphology

The fixed jejunum was cut into 0.5 cm segments. The segments was dehydrated, and embedded in paraffin. Serial sections were prepared with a thickness of 4 μm and stained with hematoxylin and eosin (H&E). Finally, the sections were sealed with neutral balsam and observed under a light microscope.

### 3.3. Protein Extraction and In-Gel Digestion

The jejunum mucosa was homogenized in phosphate buffer saline (PBS) buffer containing protease inhibitor and phosphatase inhibitor (Roche, Basel, Switzerland) using glass homogenizer. The homogenate was centrifuged at 9000× *g* for 30 min and then the supernatant was collected. Finally, the protein concentration of each sample was determined by the Bradford method (Bio-Rad, Hercules, CA, USA).

Moreover, 12 rats were divided into two groups (CON and SMG groups, 6 rats in each group). In the CON group, equal amounts of protein from each sample was taken, and then two samples were mixed into one sample. Finally, three samples (named as CON-1, CON-2 CON-3) were obtained for the CON group. The SMG samples were prepared in the same way to get SMG-1, SMG-2, and SMG-3. There were three biological replicates in each group. Each sample was prepared separately according to the following procedure. The protein samples were mixed with a 4 × loading buffer at a ratio of 3:1 and then boiled in a boiling water bath for 10 min. Samples were separated in 5% sodium dodecyl sulfate-polyacrylamide gel electrophoresis (SDS-PAGE) stacking gel (80 V for 20 min) and 12% SDS-PAGE separating gel (110 V for 80 min) in a vertical protein electrophoresis chamber. After electrophoresis, the separating gel was removed and stained with Brilliant Blue G (Solarbio, Beijing, China) for 3–4 h, and then washed with 20% methanol, repeatedly, until the strips were cleared. The stained gel was cut into strips along the lane after separation and each lane was cut into 4 parts. In-gel digestion mainly consisted of three steps: reduction, alkylation, and trypsin digestion [105]. Firstly, the gel pieces were rehydrated in 10 mM of dithiothreitol (DTT) at 60 °C for 30 min. Then, the liquid was discarded, and acetonitrile was added for 30 min to dehydrate. Secondly, the pieces of gel were soaked in the dark in 55 mM of iodoacetamide (IAM) at room temperature for 30 min to be sufficiently alkylated. After reduction and alkylation, 12.5 ng/μL of trypsin (Promega, Madison, WI, USA) was added for enzymolysis at 37 °C overnight. Acetonitrile was put into the test tube for 15 min and the liquid was collected. Finally, dry peptide powder was obtained by rapid vacuum concentration and stored at −80 °C.

### 3.4. HPLC-MS/MS Analysis

The dry peptide samples were dissolved in acetonitrile-H_2_O-formic acid (2%:98%:0.1%) solution, and 5 μL of each peptide solution was separated by one-dimensional (1D)-ultra nanoflow high performance liquid chromatography (HPLC) system (Eksigent Technologies, Silicon Valley, CA, USA) coupled to a SCIEX 4600 Q-TOF mass spectrometer (SCIEX, Boston, MA, USA). A NanoLC Trap precolumn (chormXP C_18_-CL-3 μm, 350 μm × 0.5 mm, SCIEX, Boston, MA, USA) was used for on-line desalting during peptide separation. C_18_ reverse-phase column (3 μm, 150 mm × 75 μm, Eksigent Technologies, USA) was selected as the stationary phase of HPLC. The mobile phase A was water containing 0.1% formic acid, and the mobile phase B was acetonitrile containing 0.1% formic acid. The flow rate of the mobile phase was 300 nL/min. The first 100 min of linear gradient elution were set as mobile phase A from 2% to 98%, and then the ratio of mobile phase A was maintained at 98% for 20 min until the whole chromatographic process was completed. 

The positive ionization mode was used for Q-TOF mass spectrometry detection with 2300 V capillary voltage and 175 V ion fragmentary voltage. The temperature of dry gas was set to 325 °C and the flow rate was maintained at 5 L/min. The scanning range of mass was from 300 to 1250 m/z. In centroid mode, the MS/MS spectrum scanning range was from 100 to 1250 m/z. The top 15 precursor ions with the highest abundance in each spectrum were selected for tandem mass spectrometry analysis with an active exclusion of 25 s. The fragments of protonated molecule ions were performed in auto MS/MS mode, and the acquisition of MS/MS was collected in data-dependent acquisition (DDA) mode.

### 3.5. Protein Identification and Bioinformatic Analysis

The intensity of each peptide was measured three times, and the average value was calculated. Software MaxQuant [106] (version 1.5.2.8, Max Planck Institute of Biochemistry, Munich, Bavaria, Germany) was used to search the mass data, and the protein sequence was analyzed by UniProt Database (https://www.uniprot.org/). Based on the reversed database, proteins, and peptides false discovery rate (FDR) were both less than 0.01. Carbamidomethyl (C) was fixed modification and oxidation (O) was variable. Trypsin was used as a digestive enzyme, and two cleavages were missed, at most. The mass tolerance of peptide was ±15 ppm, while mass tolerance of fragment was ±50 ppm.

The protein quantitation was performed by the Proteome Discoverer software (version 2.4, Thermo Scientific, Walsham, MA, USA) based on extracted ion chromatogram (XIC) area calculation. The ratio of protein quantitation in the SMG group to that in CON group was defined as fold change. When the value of fold change was >2 or <0.5 and the *p*-value of *t*-test was lower than 0.05, this protein was identified as upregulated or downregulated DEPs. Then, the list of DEPs was analyzed by online software. Gene-Ontology (GO) classification was performed using PANTHER (version 15.0, http://www.pantherdb.org/, Thomas lab at the University of Southern California, Los Angeles, CA, USA) to analyze the biological process (BP) and molecular function (MF) of DEPs. Moreover, the KEGG pathway was classified by DAVID (version 6.8, https://david.ncifcrf.gov/, Laboratory of Human Retrovirology and Immunoinformatics, Frederick, MD, USA).

### 3.6. Western-Blot Analysis of IDMEs

The jejunum mucosa of rats in two groups was homogenized in precooled Radio Immunoprecipitation Assay (RIPA) lysate (Roche, Basel, Switzerland) containing protease inhibitors and phosphoprotease inhibitors. The suspension was centrifuged for 10 min (4 °C, 12,000× *g*) and the supernatant was collected. Protein concentration of each sample was determined by the Bradford (Bio-Rad, Hercules, CA, USA) method. The supernatant sample was mixed with protein loading buffer and desaturated in a boiling water bath for 10 min. Proteins were separated by 12% SDS-PAGE gel followed by being transferred onto 0.22 μm polyvinylidene fluoride (PVDF) membrane (Millipore, Massachusetts, USA). The membrane was blocked in 5% of skimmed milk (BD, Lake Franklin, NJ, USA) for 2 h at room temperature, and then the membrane was incubated with corresponding primary antibodies (mouse anti-CYP1A2 monoclonal antibody, rabbit anti-CYP2D1 monoclonal antibody, rabbit anti-CYP2E1 monoclonal antibody, rabbit anti-CYP3A2 monoclonal antibody, rabbit anti-ADH1 monoclonal antibody, rabbit anti-GSTM5 monoclonal antibody; Abcam, Cambridge, UK) respectively at 4 °C overnight. The membrane was washed with Tris-Buffered Saline Tween-20 (TBST) buffer, and the washing solution was changed every 10 min for 5 times in total. Then, the membrane was incubated with horseradish peroxidase (HRP) conjugated secondary antibody (goat anti-mouse IgG, goat anti-rabbit IgG, ZSGB-Bio, Beijing, China) for 2 h at room temperature. After incubation, the membrane was washed for five times with TBST buffer in the same way as before, and then color rendered by enhanced chemiluminescence (ECL) reagent (Millipore, Billerica, MA, USA) under ChemiDoc^TM^ XRS+ (Bio-Rad, Irvine, CA, USA) software. The gray value of each strip was collected by Image Lab TM Software (version 3.0, Bio-Rad, Irvine, CA, USA). The relative expression levels of proteins were expressed by the ratio of gray value of the target band to the total proteins in the same lane [107].

### 3.7. Determination of ADH, ALDH, and GST Activity 

The activity of ADH, ALDH, and GST were determined by kits (Solarbio, Beijing, China). Experimental steps were following the kit instructions. Definitions of ADH and ALDH activity was as follows: the amount of enzyme that catalyzes the oxidation of 1 μmol NADH or the reduction of 1 nmol NAD+ per min per mg protein is one enzyme activity unit (U/min·mg prot). It was specified that 1 μmol 2,4-dinitrochlorobenzene (CDNB) catalyzed to bind to glutathione (GSH) per min per mg protein was one GST enzyme activity unit (U/min·mg prot).

### 3.8. Statistical Analysis

SPSS 20.0 software (IBM, Amonk, NY, USA) was used for statistical analysis, and the results were expressed as mean ± SD. The difference between the two groups was determined by Analysis of Variance (ANOVA). When the *p*-value was less than 0.05, it was considered to be statistically significant.

## 4. Conclusions

The results of the present study indicate that SMG could affect intestinal metabolism, including metabolism of glucose, amino acids, drugs, and xenobiotics. In addition, some intestinal metabolic enzymes related to drug metabolism have been significantly altered. These changes may affect intestinal health and disease-related intestinal homeostasis. Moreover, it may cause changes in the pharmacokinetics or pharmacodynamics of the drugs taken by astronauts, which may lead to potential toxicity or undesirable therapeutic outcomes. This article provided some preliminary information on IDMEs under microgravity. It revealed the potential effect of SMG on intestinal metabolism, which may be helpful to understand the intestinal health of astronauts and medication use.

## Figures and Tables

**Figure 1 molecules-25-04391-f001:**
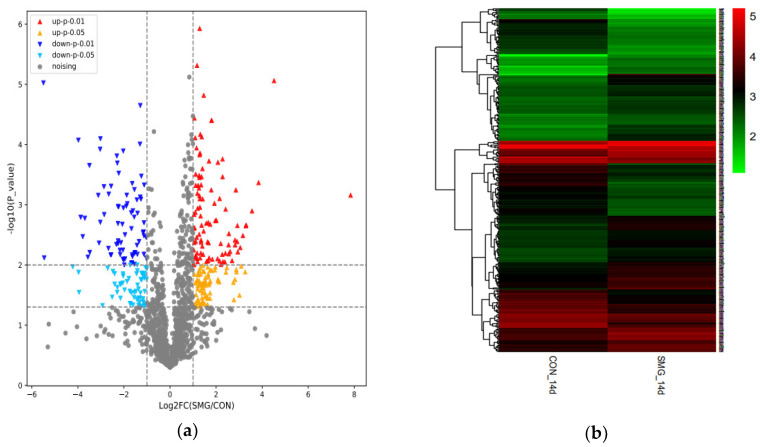
Results of all identified proteins and differentially expressed proteins, and verification by Western-blot. (**a**) Volcano plot of all identified proteins; (**b**) cluster map of all differentially expressed proteins (DEPs) after 14 d simulated microgravity (SMG); (**c**) Western-blot of alcohol dehydrogenase 1 (ADH1) and glutathione S-transferase mu 5 (GSTM5) in rat small intestine under 14 d SMG for verifying the results of MS. Compared with the control group (CON ), * *p* < 0.05.

**Figure 2 molecules-25-04391-f002:**
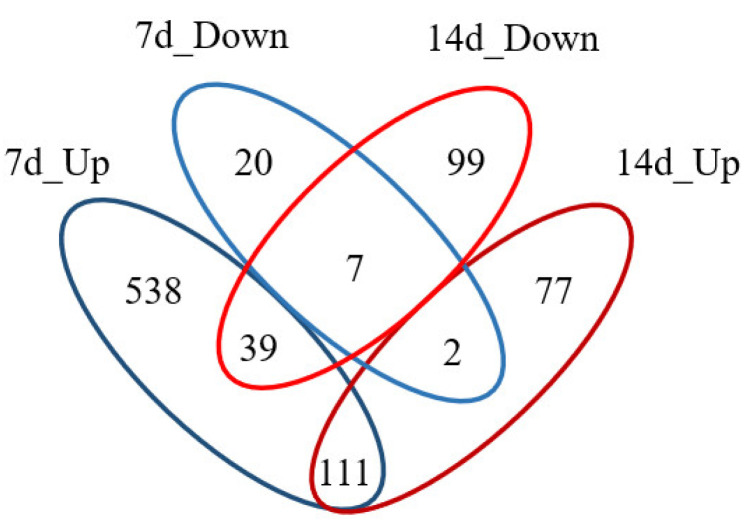
Comparison of DEPs between 14 d and 7 d SMG. From 7 d to 14 d SMG treatment, the expression of 558 DEPs in rat intestinal mucosa returned to the normal levels. Moreover, 111 DEPs were continuously upregulated, and 7 DEPs were continuously downregulated. Note: results of 7 d SMG DEPs were cited from Reference [11].

**Figure 3 molecules-25-04391-f003:**
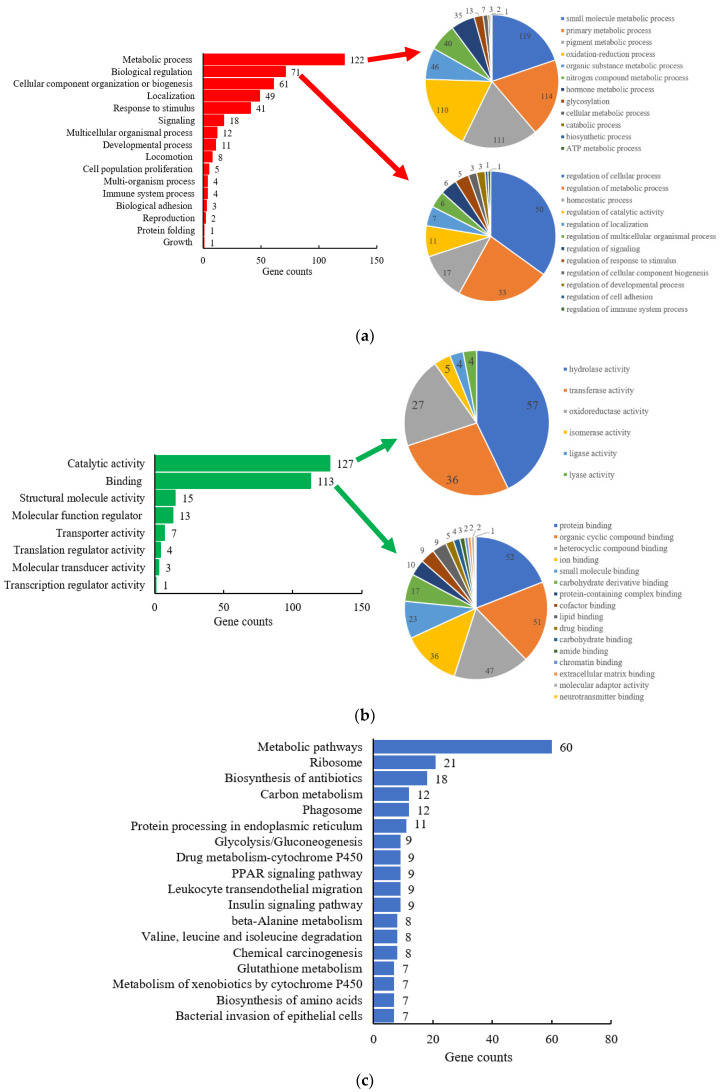
Gene-Ontology (GO) classification and Kyoto Encyclopedia of Genes and Genomes (KEGG) pathway analysis of DEPs. (**a**) Biological process (BP); (**b**) molecule function (MF); (**c**) KEGG pathway. Pie charts in (**a**,**b**) showed the details of the top two classifications in BP and MF, respectively.

**Figure 4 molecules-25-04391-f004:**
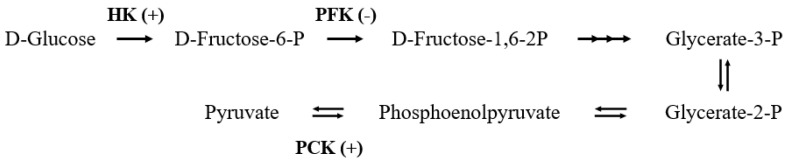
Glycolysis and gluconeogenesis processes and related DEPs. Compared with CON groups, (+) upregulated, (-) downregulated.

**Figure 5 molecules-25-04391-f005:**
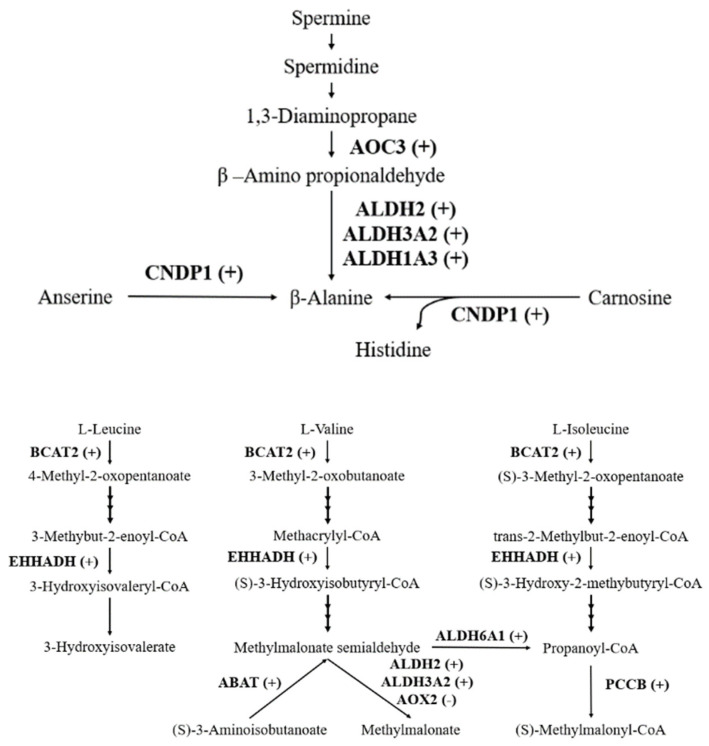
Amino acid metabolism-related DEPs. Compared with CON groups, (+) upregulated, (-) downregulated.

**Figure 6 molecules-25-04391-f006:**
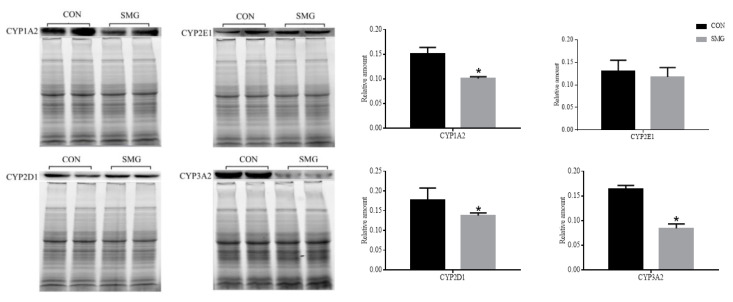
Western-blot of CYP1A2, 2E1, 2D1, 3A2 in rats’ small intestines under 14 d SMG. Compared with the CON group, * *p* < 0.05.

**Table 1 molecules-25-04391-t001:** DEPs involved in glucose metabolism.

Gene IDs	Protein Names	Fold Change	*p*-Value
Q8K4D8	Aldehyde dehydrogenase family 1 member A3 (ALDH1A3)	3.37	0.0121
P07379	Cytosolic phosphoenolpyruvate carboxykinase (PCK1)	3.08	0.0122
P27881	Hexokinase-2 (HK2)	2.58	0.0214
P05708	Hexokinase-1 (HK1)	2.40	0.0031
P11884	mitochondrial Aldehyde dehydrogenase (ALDH2)	2.33	0.0005
P30839	Aldehyde dehydrogenase family 3 member A2 (ALDH3A2)	2.17	0.0005
P30835	ATP-dependent 6-phosphofructokinase, liver type (PFK)	0.49	0.0078
P00884	Fructose-bisphosphate aldolase B (ALDOB)	0.20	0.0002
P06757	Alcohol dehydrogenase 1 (ADH1)	0.12	0.0007

**Table 2 molecules-25-04391-t002:** DEPs involved in amino acid metabolism.

Gene IDs	Protein Names	Fold Change	*p*-Value
P50554	Mitochondrial 4-aminobutyrate aminotransferase (ABAT)	8.96	0.0005
Q02253	Aldehyde dehydrogenase family 6 member A1 (ALDH6A1)	5.38	0.0031
P07633	Mitochondrial Propionyl-CoA carboxylase beta chain (PCCB)	5.33	0.0012
Q8K4D8	Aldehyde dehydrogenase family 1 member A3 (ALDH1A3)	3.37	0.0121
P07896	Peroxisomal bifunctional enzyme (EHHADH)	2.61	0.0353
O08590	Membrane primary amine oxidase (AOC3)	2.49	0.0001
P11884	mitochondrial Aldehyde dehydrogenase (ALDH2)	2.33	0.0005
Q66HG3	Beta-Ala-His dipeptidase (CNDP1)	2.33	0.0146
P30839	Aldehyde dehydrogenase family 3 member A2 (ALDH3A2)	2.17	0.0005
O35854	Mitochondrial branched-chain-amino-acid aminotransferase (BCAT2)	2.06	0.0042
Q5QE78	Aldehyde oxidase 2 (AOX2)	0.41	0.00002

**Table 3 molecules-25-04391-t003:** Results of metabolic enzyme activity determination.

Metabolic Enzymes	CON Group (U/min mg Prot)	SMG Group (U/min mg Prot)
ADH	4.75 ± 0.26	2.93 ± 0.57 *^,1^
ALDH	15.83 ± 1.82	16.30 ± 2.11
GST	20.67 ± 1.49	16.55 ± 1.17

^1^ Compared with CON group, date was expressed as mean ± SD, * *p* < 0.05.

**Table 4 molecules-25-04391-t004:** DEPs involved in drugs and xenobiotics metabolism.

Gene IDs	Protein Names	Fold Change	*p*-Value
Q9Z1B2	Glutathione S-transferase Mu 5 (GSTM5)	7.54	0.0041
P08011	Microsomal glutathione S-transferase 1 (MGST1)	4.92	0.0089
Q8K4C0	Dimethylaniline monooxygenase [N-oxide-forming] 5 (FMO5)	3.57	0.0016
Q8K4D8	Aldehyde dehydrogenase family 1 member A3 (ALDH1A3)	3.37	0.0121
P04041	Glutathione peroxidase 1 (GPX1)	2.40	0.0425
P15684	Aminopeptidase N (APN)	2.01	0.0097
Q5QE78	Aldehyde oxidase 2 (AOX2)	0.41	0.0000
P14942	Glutathione S-transferase alpha-4 (GSTA4)	0.32	0.0004
P04903	Glutathione S-transferase alpha-2 (GSTA2)	0.21	0.0003
P46418	Glutathione S-transferase alpha-5 (GSTA5)	0.14	0.0005
P06757	Alcohol dehydrogenase 1 (ADH1)	0.12	0.0007

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
