# Peer review of "Investigation on Intestinal Proteins and Drug Metabolizing Enzymes in Simulated Microgravity Rats by a Proteomics Method"

_molecules, 2020, doi:10.3390/molecules25194391_

Round 1

Reviewer 1 Report

The manuscript is dedicated to the study of an influence of simulated microgravity onto a protein profile of rats’ small intestine. The work was done on a good experimental level and provides some interesting data (proteins) for future analysis. Still, there are some points which might be improved.

  1. Reduction in activity of ALDH should be explained in more details, because according to the proteomic data expression levels of respective enzymes were up-regulated.
  2. More complete comparison of presented results with previously published by the same authors is required (reference 11 in the manuscript). Why the number of DEPs was more than 2 times lower than in the samples after 7 d? Why some proteins did or did not return to the normal level? It would be interesting to add a hierarchical clustering figure indicating DEPs between 7 and 14d SMG samples.
  3. Results of GO analysis presented on the general levels in the graphs. It will be more informative to present them with more specific ones.
  4. Histomorphology was not described in Mat&Met part. Did you use any precolumn during peptide separation? Please, provide more exact description of the gradient. 4600 Q-TOF mass spectrometer was not produced by Agilent.
  5. You wrote that IDA mode was used, but above (line 462) you wrote, that top 5 precursors were selected meaning DDA mode. Which one was applied?
  6. Please add a reference for the MaxQuant software.
  7. It’d be better also to indicate with arrows the place of respective bands on the pictures of Western-blots.
  8. Line 35: “The present study firstly provided…”. It is a questionable statement, because these data might be found in your previous paper (DOI: 10.1016/j.actaastro.2018.11.013)
  9. Lines 36-37 and 524-526: too strong statements, which should be avoided.
  10. There are no references in the text to the figures 3 and 4.
  11. Line 120: “in Figure 3 (c)”; should be “in Figure 1 (c)”
  12. Line 124: “all identified proteins”; Was it really all proteins or all DEPs?
  13. Lines 401-402: “As far as we know, there is no report about the effect of MG or SMG on the expression of CYP450 in small intestine”; can’t you get these results from your published data at 7 days?
  14. Line 428: “Protein extraction and in-gel digestion”. There is no description of protein extraction procedure in this section.
  15. Line 464: “the activity was eliminated for 25 s”; what activity do you mean?
  16. Quality of language must be improved, a grammar review/adjustment must be completed: there are a lot of mistakes of different kinds. Some of them (not all) are marked in the attached file.

Reviewer 2 Report

The authors sought to characterize the change of intestinal mucosa proteins in response to intestinal drug metabolizing enzymes. To discover the differentially expressed proteins associated with simulated microgravity, label-free quantitative proteomics studies were carried out using MS proteomics results were supported by Western-blot (WB) analysis. As the result, they identified a significant number of proteins, and discovered grouped proteins were significantly altered in rat. Pathways analysis was performed. I acknowledged that a lot of effort was contributed to achieve this goal. Nevertheless, there are some shortcomings needed to be corrected before being published in the journal. One major concern is the quantification of WB. Authors claimed that “the relative expression levels of proteins were expressed as the ratio of gray value of the target band to that of total proteins”. The intensity of total protein is not reliable. I would suggest the quantification to be performed against loading control (e.g. GAPDH or actin)- single band. Also, authors should deposit the MS raw data to ProteomeXchange Consortium.    

Round 2

Reviewer 1 Report

From scientific viewpoint I do not have any further comments or corrections. Still, there is a problem with language. The main obstacle in clear understanding of written text is a style. I would recommend to send the paper for a professional editing.

Lines 331 and 361: you referred to Table 3, should be Table 4.

Please, add to the paper information where the raw proteomic data are available.

Reviewer 2 Report

All concerns have been addressed. 

Author Response

Dear reviewer,

     Thank you for reviewing our manuscript and valuable suggestions. The manuscript has been revised carefully.

Sincerely,

Yujuan Li

This manuscript is a resubmission of an earlier submission. The following is a list of the peer review reports and author responses from that submission.

Round 1

Reviewer 1 Report

The manuscript by Liu et al describes changes in the proteome of the intestinal mucosa in a rat model of micro-gravity in which rats were tail-suspended to make their hind limbs off the ground for 14 days, in comparison to a control group of normally moving rats.

The topic is potentially interesting due to the scarcity of data about the physiological effects of microgravity but the experimental design presents very important flaws that make me recommend the manuscript to be Rejected.

The most important criticism is related to how samples are managed. As described in Materials and Methods, 12 rats were randomly divided into two groups (Control and microgravity (SMG)) and equal amounts of protein from each individual were mixed for each group.

That means that only one pool of samples for each group has been examined. This does not have any statistical validity. At least two groups for each CON and SMG conditions, coming from randomly pooling half of the samples in each condition, had to be construed and compared.

Not even technical replicates have been made.

Other points:

It will be most interesting to have plasma samples from these rats and to analyse whether changes in glucose uptake and metabolism in the intestinal mucosa. In any case, the sentence “It can be concluded that SMG may lead to disorder of glucose metabolism, which may then 209 induce polydipsia, polyphagia, polyuria, weight loss and other symptoms related to abnormal 210 glucose metabolism” is too speculative with only the proteomic data from the intestine. It would be maybe acceptable with supporting data from plasma and/or liver.

A network analysis with all differentially abundant proteins should be performed on top of the GO analysis to ascertain which are the pathways most affected by microgravity.

In the supplementary table, many proteins appear to have higher fold-changes that the ones chosen by the authors (kinesin 230-fold change). I understand that the authors choose the metabolism-related proteins after the GO analysis. This is one of the reasons why a network analysis must be carried out.

A morphological analysis of the mucosa to complement the proteomic data must be performed.

One or more diagrams to show the relationship of the different proteins in the same biochemical pathway would be convenient to have a global vision of the foreseen glucose and aminoacid metabolism.

Anminoacid metabolism: several of the identified proteins are related to amino acid transport from the lumen to the blood through the mucosa, more than with amino acid metabolism in the intestine. That must be taken into account in the discussion.

Some of the sentences are too much speculative and out of context in the absence of supporting results, especially those related to the central nervous system (GABA in lines 239-241 and 278-283)

Other, less important, criticisms have been found, that may be dealt with in a new submission of the manuscript.

Reviewer 2 Report

This is a nice paper on actual hot topic. I’m satisfied with results and methods which authors apply for proteomic study. The author succeed to discover and validate the  differentially expressed proteins (DEPs) including 11 DEPs which were involved in exogenous drug and xenobiotics metabolism. I recommend this paper for publishing in Molecules. There are only few minor comments listed below:

  • It worth mentioning how the rat gut differs from the human gut and what limitations this imposes on the interpretation of results in relation to humans. Or explain why it is good to use rats as a model, rather than other laboratory animals.
  • It is worth mentioning the work on the study of enzymes involved in the metabolism of xenobiotics in rats after Spacelab Life Sciences mission 1, SLS-1 [Rabot S, Szylit O]. in SLS-1 there was an induction of intestinal glutathione-S-transferase. If possible, explain the reason for the diverging results. [Rabot S, Szylit O, Nugon-Baudon L, Meslin JC, Vaissade P, Popot F, Viso M.Variations in digestive physiology of rats after short duration flights aboard the US space shuttle. Dig Dis Sci. 2000 Sep;45(9): 1687-95. doi: 10.1023/a: 1005508532629.]

There are typos in the text:

line 105: was down-regulation or was down-regulated? was up-regulation or was up-regulated?

line 94: IDEMs - need a transcript, or is it a typo and means IDMEs?

line 327: the meaning of the word defeminated is not clear. Maybe you mean "determined"?

Authors should carefully correct the text of the article again.

Reviewer 3 Report

The manuscript describes the proteomics analysis of intestinal proteins in rats subjected to microgravity. The proteomics approach is a standard bottom-up technique and the quantification is done in a label-free manner. The authors identify several differentially regulated proteins and further validate their findings using western blots and enzymatic assays. The authors further discuss in an extensive manner the relevance of the identified proteins in the context of microgravity and aerospace technologies. 

The manuscript is generally well organized. The data reported has one major issue - please see comment 2 and the methods need extensive improvements - please see comments 3. The English language needs work, there are many spelling and phrasing mistakes that make the text difficult to follow and understand. 

Major comments:

  1. Methods section mentions a total of 12 rats for this study. Please clearly state the number of biological replicates for the proteomics experiments (ie. the number of rats for CON and SMG groups that were actually used for proteomics experiments).  
  2. It is unclear for me the reason behind mixing biological replicates before gel separation and proteomics analysis. Usually, biological replicates are treated separately for better statistics. How are the authors sure that the amounts of proteins were mixed in equal amounts? How do the authors deal with the rats' individual variability?
  3. Please provide more details on how the quantitation of the proteins was performed. The authors state that "The intensity of each protein...was measured". An MS/MS approach does not measure proteins, but peptides (precursor ions). Also, label-free cuantitation can be done in several ways (spectral counting, absolute or relative,  or XIC quantitation). 

Minor comments:
1. figures 1 to 3 can be merged into a single figure with 4 parts. 

2. figure 4 - the two parts can be aligned horizontally and eventually merged with figure 5.

3. the words "and so on" are repeated many times across the manuscript leading to ambiguity. 

Round 2

Reviewer 3 Report

The authors have put some efforts to improve the manuscript tacking into account comments from all the reviwers. Still, as it is the manuscript text does not answer my major comment related to sample processing. 

The authors now state that: "12 rats were divided into two groups (CON and SMG group, 6 rats in each group). Protein level in each sample was determined by Bradford Assay Kit. In CON group, equal amounts of protein from each sample was taken, and then two samples were mixed into one sample"

Please clearly define what you consider to be a sample. One proteomics sample equals one animal or not? As it is described, I can understand that proteins extract from two animals were mixed to give one proteomics sample that was further processed. If so, why did the authors did it this way? Excluding the economical reasons (less instrument time, 3 vs 6 samples to process), did the authors had any other reasons for not treating each animal as a sample and running the proteomics analysis for true biological replicates?